# Low Childhood Nature Exposure is Associated with Worse Mental Health in Adulthood

**DOI:** 10.3390/ijerph16101809

**Published:** 2019-05-22

**Authors:** Myriam Preuß, Mark Nieuwenhuijsen, Sandra Marquez, Marta Cirach, Payam Dadvand, Margarita Triguero-Mas, Christopher Gidlow, Regina Grazuleviciene, Hanneke Kruize, Wilma Zijlema

**Affiliations:** 1Faculty of Health, Medicine and Life Sciences, Maastricht University, Minderbroedersberg 4-6, 6211 LK Maastricht, The Netherlands; m.preuss@student.maastrichtuniversity.nl; 2Barcelona Institute for Global Health (ISGlobal), Doctor Aiguader 88, 08003 Barcelona, Spain; mark.nieuwenhuijsen@isglobal.org (M.N.); sandra.marquez@isglobal.org (S.M.); marta.cirach@isglobal.org (M.C.); payam.dadvand@isglobal.org (P.D.); margarita.triguero@isglobal.org (M.T.-M.); 3Universitat Pompeu Fabra (UPF), Doctor Aiguader 88, 08003 Barcelona, Spain; 4CIBER Epidemiología y Salud Pública (CIBERESP), Melchor Fernández Almagro 3-5, 28029 Madrid, Spain; 5Centre for Health and Development, Staffordshire University, Leek Road, Stoke-on-Trent ST4 2DF, UK; C.Gidlow@staffs.ac.uk; 6Department of Environmental Sciences, Vytautas Magnus University, K. Donelaicio str. 58, 44248 Kaunas, Lithuania; Regina.grazuleviciene@vdu.lt; 7Center for Sustainability, Environment and Health, National Institute for Public Health and the Environment (RIVM), Antonie van Leeuwenhoeklaan 9, 3721 MA Bilthoven, The Netherlands; hanneke.kruize@rivm.nl

**Keywords:** childhood nature exposure, natural outdoor environments, nature perception, mental health, vitality, greenness, green space

## Abstract

Exposure to natural outdoor environments (NOE) is associated with health benefits; however, evidence on the impact of NOE exposure during childhood on mental health (MH) and vitality in adulthood is scarce. This study was based on questionnaire data collected from 3585 participants, aged 18–75, in the PHENOTYPE project (2013) in four European cities. Mixed models were used to investigate associations between childhood NOE exposure and (i) MH; (ii) vitality (perceived level of energy and fatigue); and (iii) potential mediation by perceived amount, use, satisfaction, importance of NOE, and residential surrounding greenness, using pooled and city-level data. Adults with low levels of childhood NOE exposure had, when compared to adults with high levels of childhood NOE exposure, significantly worse mental health (coef. −4.13; 95% CI −5.52, −2.74). Childhood NOE exposure was not associated with vitality. Low levels of childhood NOE exposure were associated with lower importance of NOE (OR 0.81; 95% CI 0.66, 0.98) in adulthood. The association with perceived amount of NOE differed between cities. We found no evidence for mediation. Childhood NOE exposure might be associated with mental well-being in adulthood. Further studies are needed to confirm these findings and to identify mechanisms underlying long-term benefits of childhood NOE exposure.

## 1. Introduction

The global burden of mental illnesses has increased over the years [1,2]. Mental health can be negatively influenced by urban built environments, including among other characteristics, exposure to noise, crowds and lack of green spaces [3,4,5,6]. Previous studies found that green spaces positively influence physical and mental health as well as general well-being in adults [7,8,9,10,11,12,13]. Living close to natural outdoor environments (NOE) and attractiveness of nearby NOE are both associated with increased recreational activity and time spent in nature [14,15,16,17,18]. The health benefits of nature have been attributed to reduction in air pollution; increased physical activity, including walking and biking; improved mental restoration through alleviation of anxiety and stress; and the beneficial effects of social interaction by reducing social loneliness and generation of social capital [19,20,21,22].

In children, indoor activities and sedentary lifestyle have increased over the years and have been associated with unfavorable behavioral conduct, lower self-esteem, poor concentration and reduced psychological well-being, quality of life, and physical health [23,24,25,26,27]. Psychological problems during childhood and adolescence can have profound long-term consequences, including poorer mental and physical health, higher levels of criminal behavior, and worse economic prospects in adulthood [28,29,30]. In contrast, the availability of NOE and the time that children spend in them have been associated with increased self-esteem, quality of life, respiratory health, physical activity, and lower body mass index (BMI) [31,32,33,34]. It has also been suggested that NOE provide benefits that directly influence cognitive development in children, as NOE provide opportunities that stimulate discovery, creativity, mastery, engagement, risk taking behavior, basic emotional states (e.g., surprise), and psychological restoration [35,36,37]. Ways in which NOE are suggested to exert indirect beneficial effects on the cognitive development of children include the mitigation of traffic-related air pollution, reduction of noise, and increased levels of physical activity [11,35,38,39,40,41]. Nature exposure during childhood might thus have long-term health implications. A nationwide study using register data in Denmark observed associations between lower residential surrounding greenness and higher risks for a range of psychiatric disorders in adulthood [42]. A study conducted in the USA found an association between higher residential surrounding greenness during childhood and lower risk of depressive symptoms in adulthood [43].

Previous studies also reported an association between time spent in green spaces in adulthood and higher levels of vitality [44]. Associations have been reported for the amount of green spaces in the neighborhood and vitality, perceived general health, and greater quality of life in adults [45,46,47]. However to our knowledge, the association between childhood NOE exposure and vitality in adulthood has not yet been analyzed.

Some authors have suggested that childhood NOE exposure can have a long-term influence upon the perception, evaluation and use of NOE, but evidence on this relationship is scarce. Nature experience during childhood has been positively associated with the development of a nature-oriented attitude and increased preference for nature-based activities in adulthood [44,48,49,50,51]. An epidemiological study in the United Kingdom found that those who frequently visited nature during childhood also made frequent nature visits in adulthood [49]. Others have found that the association between purposeful nature visits and mental health was stronger for adults with less childhood NOE exposure than for adults with more childhood NOE exposure [44]. These findings show that childhood NOE exposure could potentially influence the way people interact with NOE in adulthood. Childhood NOE exposure might thus affect the choice for living in an area with a certain amount of residential surrounding greenness, and the perceived amount, use, satisfaction, and importance of NOE in adulthood. These factors could subsequently lead to health benefits and mediate the association between childhood NOE exposure and mental health or vitality. Therefore, it is important to consider these variables as potential mediators in the analysis.

It can be concluded that existing literature highlights the importance of childhood NOE exposure for physical, mental, and cognitive development. High levels of time spent in NOE in adulthood are associated with favorable mental health and vitality states in adulthood. Although to our knowledge, few studies have explored the association between childhood NOE exposure and mental health and vitality in adulthood. Additionally, there is limited information on the association between childhood NOE exposure and residential surrounding greenness, perceived amount, use, satisfaction, and importance of NOE in adulthood, as well as the potential mediating role of these factors on the association between childhood NOE exposure, mental health and vitality.

We hypothesized that low levels of childhood NOE exposure would be associated with the choice of a neighborhood exhibiting lower residential surrounding greenness, lower perceived amount of NOE, lower use of NOE, lower satisfaction with NOE, lower importance of NOE, and lower mental health and vitality scores in adulthood when compared to adults with high levels of childhood NOE exposure. Our study aimed to assess the association between childhood NOE exposure and mental health and vitality in adulthood. As a secondary aim, we evaluated the mediating role of residential surrounding greenness, perceived amount, use, satisfaction, and importance of NOE in adulthood as potential mechanisms underlying this association. We hypothesized that these NOE indicators would positively influence the association between childhood NOE exposure and the outcomes mental health and vitality.

## 2. Methods

### 2.1. Ethical Approval

The appropriate ethical committees of the involved research institutes approved data collection in conjunction with the “Positive health effects of the natural outdoor environment in typical populations in different regions in Europe” (PHENOTYPE) project: Parc de Salut Mar, Clinical Research Ethics Committee reference number 2011/4206/I; Medisch Ethische Toetsingscommissie UMCU reference number 12/595; Lietuvos Bioetikos Komitetas reference number 6B-12-147; Faculty of Health Sciences’ Ethics Panel, no reference number). Written informed consent was obtained from all participants [44,52]. The study was conducted in accordance with the Declaration of Helsinki [53].

### 2.2. Study Setting and Data Collection

This cross-sectional study was based on data from the PHENOTYPE project. Data were collected from a representative, random sample of residents in different neighborhoods of four European cities: Barcelona (Spain), Doetinchem (the Netherlands), Kaunas (Lithuania), and Stoke-on-Trent (UK). Data collection took place between May and November 2013. The questionnaire was first developed in English and was then translated into Dutch, Spanish, Catalan, and Lithuanian. In all cities except for Kaunas, information was collected by interview-administered questionnaires. In Kaunas, postal questionnaires were sent to the participants [44,52]. Further information on the background of the study and data collection can be found in the PHENOTYPE study protocol [52].

### 2.3. Study Population

In each city, data were collected from 1000 randomly selected adult residents (aged 18 to 75 years), across approximately 30 different spatial units, which were selected to ensure variability in access to NOE (distance to NOE) and diversity in socioeconomic status (SES). In total, data from 3986 participants were available, although 401 were excluded due to missing data on childhood NOE exposure, age, gender, education, perceived financial situation, SES of the neighborhood, household composition, or smoking status. This provided a final sample of 3585 participants for analysis, with an equal distribution among the four cities (Table 1, Table 2, Table 3).

### 2.4. Study Data

Natural outdoor environments were defined as all public and private open spaces that contain “green” and/or “blue” natural elements. Green natural elements included, among others, roof gardens, forests, farmland, city parks, and nature reserves. Blue spaces encompassed water structures such as canals, ponds, creeks, rivers, lakes, and beaches [52]. 

*Childhood NOE exposure:* Participants were asked to retrospectively rate how often they spent time in NOE during their childhood. This included purposeful visits (e.g., hiking in natural parks with the parents) and non-purposeful visits (e.g., playing outside in the backyard). Answers were scored on a five-point scale, which was dichotomized (“never”, “sometimes” as low levels of childhood NOE exposure; “regularly”, “often”, “very often” as high levels of childhood NOE exposure) as done in previous studies [44].

*Socio-demographic characteristics*: Age (in years), gender (male; female), birth in the country of residence (yes; no), smoking status (current; former; never), household composition (living alone; with partner but without children; with children <12 years; with children ≥12 years; other), educational achievement (primary school or no education; secondary school or further education; university degree or higher), perceived income situation (cannot make ends meet; just have enough to get along; comfortable), and SES of the neighborhood were included in the adjusted models as potential confounders. 

*Perceived amount of NOE:* Respondents were asked to describe their close living environment in terms of green and blue. The question consisted of three items concerning the participants’ (a) street, (b) neighborhood, and (c) view from home, and each item was answered on a five-point scale ranging from “not at all” to “very”. A summary score was calculated with higher scores indicating higher perceived amount of NOE.

*Use of NOE*: It was assessed by combining two associated questions concerning the visits (frequency and duration per visit) of NOE close to home during the last four weeks. Answers were measured on five-point scales (Frequency: “n.a.”, “never”, “≤1”, “2–3”, “1–4 times weekly”, “(almost) daily; Duration: “n.a.”, “<1 hour [h]”, “1–2 h”, “3–5 h”, “6–10 h”). The middle values were determined (e.g., 3–5 h equaling 4) and the values of both questions were multiplied with each other to calculate the total hours spent in NOE during the last four weeks. 

*Residential surrounding greenness:* This was assessed by using the normalized difference vegetation index (NDVI), derived from satellite images from Landsat 8 at a resolution of 30 m × 30 m [54]. The NDVI is based on the fact that healthy vegetation absorbs most visible light and at the same time reflects large parts of near-infrared light. It provides a number on a scale from −1 to +1, with higher values indicating higher density of green vegetation [54]. Average NDVI values were calculated within a (Euclidean) buffer of 100 m, 300 m, and 500 m around participants’ dwellings [55].

*Satisfaction with NOE*: The satisfaction with NOE in the surrounding living environment was assessed in terms of (a) quality, (b) amount, (c) maintenance, and (d) safety of the green/blue environment. Answers were scored on a five-point scale (“very dissatisfied” to “very satisfied”), with higher scores indicating higher satisfaction. A summary score of the four items was calculated by summing the points of all items together.

*Importance of NOE:* Participants were asked how important green/blue spaces near place of residence are for: (a) physical activity, (b) social activities, (c) relaxation, and (d) that there are green walking and biking paths to go to work, school and other destinations. Answers were measured on a five-point scale (“not important at all” to “very important”). Higher scores indicated higher importance followed by calculation of a summary score. 

*Mental health and vitality:* Mental health and vitality were assessed using two subscales from “The Medical Outcome Study 36-item Short-Form Health Survey (SF-36)”, which have previously been shown to be valid and reliable [56,57]. The five-item mental health component assessed nervousness and feelings of depression during the past four weeks. The four-item vitality component concerned perceived level of energy and fatigue and is recommended in order to better capture differences in subjective well-being [56]. All nine items were scored on a six-point scale, ranging from “All of the time” to “none of the time”. In line with previous studies, two items of the mental health and two of the vitality subscale were reverse-coded [10]. In case a maximum of two items of the mental health subscale or a maximum of one item of the vitality subscale were missing, they were replaced by the average of the remaining items. By doing so, we followed the manual and interpretation guide for the SF-36 Health Survey [57]. Accordingly, sum scores were calculated and transformed into continuous scales (0–100), with higher scores reflecting better mental health or vitality [10,56,57].

### 2.5. Statistical Analysis

Equivalence tests with Benjamini–Hochberg adjustment for false discovery rates (5%) were used to test for differences between low and high level childhood NOE exposure groups [58]. We analyzed the association between childhood NOE exposure and mental health/vitality as the main aim of the study. Then, following the steps as suggested by Baron and Kenny, we investigated the mediation roles of residential surrounding greenness, perceived amount, use, satisfaction, and importance of NOE in the association between childhood NOE exposure and mental health as well as vitality (see step 2 and 3 below) [59]. All analyses were performed with pooled data and for the four cities separately. Random intercepts were used. In the analyses with pooled data, we included city and neighborhood as random intercepts. In the analyses per city, we only included neighborhood as a random intercept. For all analyses, first crude models were built, which were then adjusted for potential confounders (age, gender, native/foreign born, level of education, perceived income situation, neighborhood SES, household composition, smoking status). All analyses were performed using Stata 15 [60].

*Step 1—The association between childhood NOE exposure and mental health/vitality:* The associations between the exposure variable childhood NOE exposure and the outcomes mental health and vitality were assessed using mixed effects regression analyses.

*Step 2—The association between childhood NOE exposure and the potential mediators:* Logistic mixed effects models were developed for the exposure variable childhood NOE exposure and the potential mediators (perceived amount of NOE, use of NOE, satisfaction with NOE, importance of NOE, and residential surrounding greenness). In the pooled analyses, the mediators were dichotomized based on the overall median. For analyses on the city level, city-specific cut points, based on the city-specific median, were used. The lower value was defined as the reference group and indicated lower rating.

*Intermediate step—The association between the potential mediators and mental health/vitality:* The associations between the potential mediators and mental health as well as vitality were assessed using mixed effects models.

*Step 3—The association between childhood NOE exposure and mental health/vitality including the potential mediators:* The potential mediators were added (separately and altogether) to the adjusted main model which has been described in step 1, in order to assess the association between childhood NOE exposure and mental health/vitality controlled for the potential confounders and for the potential mediators.

### 2.6. Sensitivity Analyses

The associations between childhood NOE exposure and mental health/vitality were also adjusted for the average residential surrounding greenness within 100 m and 500 m buffers to assess sensitivity to different buffer sizes. Analysis specified in step 3 (the association between childhood NOE exposure and mental health/vitality including the potential mediators) was also conducted with a different cut off for childhood NOE exposure: “never”, “sometimes,” “regularly” as low levels of childhood NOE exposure (*n* = 1369, 38.2%); “often”, “very often” as high levels of childhood NOE exposure (*n* = 2216, 61.8%), to assess sensitivity to different cut offs of childhood NOE exposure.

## 3. Results

### 3.1. Study Population Characteristics 

The mean age of the study population was 51.7 years (SD 15.8), 54.9% of the study population was female, 15.7% of the participants had low and 84.3% had high levels of childhood NOE exposure. The group with low levels of childhood NOE exposure was largest in Stoke-on-Trent (25.4%) and smallest in Doetinchem (7.1%) (Table 1, Table 2, Table 3). A higher proportion of participants with low levels of childhood NOE exposure were born in the country of residence (93.1% vs. 89.7%; *p* = 0.014) and had achieved less often the highest educational degree (42.2% vs. 50.6%; *p* < 0.001) when compared to participants with high levels of childhood NOE exposure (Table 4, Table 5, Table 6).

### 3.2. The Association between Childhood NOE Exposure and Mental Health/Vitality

Compared to high levels of childhood NOE exposure, low levels of childhood NOE exposure were significantly associated with lower mental health scores in adulthood in the pooled analyses (Coef. −4.13; 95% CI −5.52, −2.74) and results were mostly consistent when stratified by city. Negative, non-significant associations were seen for childhood NOE exposure and vitality in the pooled analyses, but a significant negative association was observed for participants in Kaunas (Table 7).

### 3.3. The Association between Childhood NOE Exposure and the Potential Mediators

Pooled analysis showed that participants with low levels of childhood NOE exposure had a significantly higher perceived amount of NOE when compared to participants with high levels of childhood NOE exposure (OR 1.32; 95% CI 1.06, 1.64), despite the results per city being mixed. No associations were found between childhood NOE exposure and use of, or satisfaction with NOE in the pooled analyses. Overall, participants with low levels of childhood NOE exposure rated the importance of NOE significantly lower than participants with high levels of childhood NOE exposure (OR 0.81; 95% CI 0.66, 0.98). No associations were found between childhood NOE exposure and residential surrounding greenness (Table 8). 

### 3.4. The Association between the Potential Mediators and Mental Health/Vitality

In the pooled analyses, no association was found between perceived amount of NOE and mental health. High use of NOE, when compared to low use, was significantly associated with higher mental health scores in the pooled analyses (coef. 2.10; 95% CI 1.07, 3.13), and in most of the analyses per city. High satisfaction with NOE, when compared to low satisfaction, was significantly associated with higher mental health scores in the pooled analyses (coef. 2.12; 95% CI 1.07, 3.17) and these associations were consistent when stratified by city. Furthermore, high importance was associated with higher mental health scores on pooled level (coef. 2.17; 95% CI 1.15, 3.19) and this finding was consistent for most of the city analyses. No associations were observed between residential surrounding greenness (within 300 m from home) and mental health (Appendix A).

Overall, participants with a high perceived amount of NOE had significantly higher vitality scores when compared to participants with low perceived amount of NOE (coef. 1.76; 95% CI 0.49, 3.02). High use of NOE, when compared to low use, was associated with higher vitality scores on pooled level (coef. 2.92; 95% CI 1.73, 4.11) and in all cities. High satisfaction, when compared to low satisfaction with NOE, was significantly associated with higher levels of vitality on pooled level (coef. 3.47; 95% CI 2.26, 4.67), and for participants in Barcelona and Kaunas. Statistically significant associations were observed between high importance of NOE and higher levels of vitality on pooled level (coef. 2.74; 95% CI 1.57, 3.92), and in the samples from Barcelona and Stoke-on-Trent. No associations were observed between residential surrounding greenness within 300 m and vitality (Appendix A).

### 3.5. The Association between Childhood NOE Exposure and Mental Health/Vitality Including the Potential Mediators

When the potential mediators were added separately to the adjusted model for mental health, all associations that were significant in the adjusted main models (Table 7) remained significant. Adding the potential mediators separately as well as altogether to the model only strengthened the associations between childhood NOE exposure and mental health in the pooled analyses, contrary to what should be found if the explored variables were mediators of the association (Table 9, Table 10, Appendix A).

The association between childhood NOE exposure and vitality in the adjusted main model was significant only for participants in Kaunas (Table 7). This association remained significant after all additional adjustments (Table 9, Table 10, Appendix A).

### 3.6. Sensitivity Analyses

The associations between childhood NOE exposure and mental health/vitality were adjusted for average residential surrounding greenness within 100 m and 500 m buffers. The results were comparable to the associations adjusted for average residential surrounding greenness within 300 m buffers (Appendix A).

The associations between childhood NOE exposure and mental health including the mediators while using a different cut off for childhood NOE exposure (“never”, “sometimes”, “regularly” as low levels; “often”, “very often” as high levels) were attenuated in the pooled analyses, but low levels of childhood NOE exposure were still significantly associated with lower mental health scores, compared with high levels of childhood NOE exposure (Appendix A). Analysis with a different childhood NOE exposure cut off resulted for Barcelona participants in associations between childhood NOE exposure and mental health that were no longer statistically significant. However, for Doetinchem participants, associations between childhood NOE exposure and mental health became stronger and statistically significant. The associations between childhood NOE exposure and vitality including the mediators while using a different cut off for childhood NOE exposure did not change the conclusions (Appendix A). 

## 4. Discussion

### 4.1. Main Findings

This is one of the first epidemiological studies to show that low levels of childhood NOE exposure are associated with worse mental health in adulthood when compared to those who had high levels of childhood NOE exposure. Childhood NOE exposure was not associated with vitality in adulthood. We hypothesized that childhood NOE exposure could shape how NOE are perceived and used, how satisfied adults are with the NOE, and how important NOE are to them, and that this could subsequently confer health benefits. We found that adults with low levels of childhood NOE exposure, when compared to adults with high levels of childhood NOE exposure, considered NOE as less important, but that the association with perceived amount of NOE varied between cities. The use of NOE, satisfaction with NOE, and importance of NOE were positively associated with higher mental health scores. These three variables and the perceived amount of NOE were also positively associated with vitality. However, neither of them acted as mediators between childhood NOE exposure and mental health.

### 4.2. Available Evidence and Potential Underlying Mechanisms

Our study revealed that low levels of childhood NOE exposure, when compared to high levels of childhood NOE exposure, were significantly associated with lower mental health scores in adulthood. These results are consistent with a study conducted by Engemann et al. who showed that children growing up with little residential surrounding greenness had higher risks of psychiatric disorders in adulthood compared to children growing up with higher amounts of greenness [42]. Maas et al. also found that certain disease clusters were more apparent in children in areas with little green spaces, with the strongest relationship for depression [61]. Bezold et al. found an association between higher residential surrounding greenness during childhood and lower risk of depressive symptoms among adults [43]. Engemann et al. also observed the strongest associations for lack of greenness and neurotic, stress-related, depressive and somatic disorders, leading them to believe that the restorative ability of greenness might explain their findings [42]. They further hypothesized that the strong associations for substance abuse disorders could mean that greenness contributes to the development of better stress-processing ability, resulting in lower needs for self-medication in later life [42]. Dadvand et al. found that exposure to NOE provides benefits for the cognitive development in schoolchildren [35]. In a subsequent study they found a positive association between lifelong exposure to greenness and increased white matter volume in various brain regions as well as increased grey matter volume in the left premotor cortex, and in the left and right prefrontal cortex at age eight [62]. Goodkind et al. found grey matter loss in psychiatric patients in the prefrontal cortex and other studies suggested that the prefrontal cortex plays a major role in depression [62,63,64]. In contrast to the three cities where associations were observed, we found no association between childhood NOE exposure and mental health for participants from Doetinchem at the city level. Doetinchem is the greenest city out of those studied, and childhood NOE exposure was high. Low variability in NOE exposure among these participants is the most likely explanation for the missing association. 

Available data and information on the association between childhood NOE exposure and vitality is scarce. Our study revealed a non-significant association for low levels of childhood NOE exposure, when compared to high levels of childhood NOE exposure, and lower vitality scores. Van den Berg et al. found that childhood NOE exposure did not interact with the association between green space visits in adulthood and vitality in adulthood [44]. A possible explanation is that feelings of vitality are not as long-term influenced as mental health and rather depend on recently undertaken restoration and regeneration measures. Another possible explanation for this finding is that the SF-36 vitality scale comprises components that refer to physical health. Therefore, it might be more strongly related to a currently active lifestyle than the SF-36 mental health scale [65].

The mediation analysis revealed that low levels of childhood NOE exposure were associated with a higher perceived amount of NOE when compared to high levels of childhood NOE exposure. This was contrary to the initial hypothesis. A potential explanation could be that that those with high levels of childhood NOE exposure have become less sensitive to the amount of NOE around them, as they had regular contact with NOE during their early life. This theory would also provide an explanation for the findings that participants with low levels of childhood NOE exposure rated the importance of NOE lower when compared to participants with high levels of childhood NOE exposure. Participants with low levels of childhood NOE exposure might be used to little contact with nature, while participants with high levels of childhood NOE exposure might take NOE for granted. Similar findings were also reported by van den Berg et al. who found that the mental health benefits gained from NOE visits in adulthood were stronger for people with low levels of childhood NOE exposure than for people with high levels of childhood NOE exposure [44].

Further, little is known about how childhood NOE exposure influences residing in green neighborhoods and perceived amount, use, satisfaction, and importance of NOE in adulthood. Previous studies reported associations between surrounding greenness as well as use of NOE and mental health in adulthood, which is why we included these as potential mediators [10,66]. We found that high use of NOE, high satisfaction with NOE and high importance of NOE were associated with better mental health. Our results also showed that high perceived amount of NOE, high use of NOE, high satisfaction with NOE, and high importance of NOE were associated with higher levels of vitality. However, these variables did not change the association between childhood NOE exposure and mental health. These findings suggest that childhood NOE exposure is independently associated with mental health in adulthood, and that it is not mediated by current residential surrounding greenness, and perceived amount, use, satisfaction, or importance of NOE. It also showed that the association between childhood NOE exposure and mental health was not confounded by current NOE exposure in adulthood. This was an important addition to a previous study that demonstrated associations between childhood NOE exposure and mental health in adulthood, but that did not take the NOE exposure in adulthood into account [43].

### 4.3. Strengths and Limitations of the Study

Our cross-sectional study, by design, had a limited capability to establish a causal relationship. Moreover, our exposure variable (childhood nature exposure) was assessed with only one question and, as it was measured retrospectively, there is the potential for recall bias [67]. A longitudinal study design or the additional assessment of childhood NOE by the participants’ parents or siblings would have increased the reliability of data on childhood NOE [42]. Additionally, information on the housing situation (high-rise vs. houses, with or without garden) and information on the family history of mental health issues would have provided a better picture of the childhood situation and it would have been favorable to control for these variables. Missing data in items of the mental health and vitality scales were replaced as recommended in the manual and interpretation guide [57]. Participants with missing data of the exposure variable or of potential confounders (*n* = 401) were excluded. Exclusion typically involves a loss of statistical power, can introduce bias in the estimates and can potentially lead to invalid conclusions [68]. Furthermore, mental health and vitality were self-reported, albeit with well-validated questionnaires. It would have been favorable to objectively measure mental health and vitality by a health professional. In addition, it would have been preferable to use GPS to track participants’ use of NOE. Also, we did not explicitly take micro- or macro-climatic differences and differences in vegetation types between the cities into account. These may have partly influenced the results, however, by using a mixed effects model with city as a random intercept, we have, to a large extent, accounted for these differences [69,70].

Our study did also have a number of strengths. Data were gathered from a large, diverse study population from different European countries with distinct cultures, including varying degrees of independent mobility of children (e.g., visits of NOE around home without parental accompaniment), valuation attached to children’s contact with nature, and perceived barriers as well as safety concerns reported by adults to make use of NOE [71,72,73]. Many previous studies on environment, nature and health have been conducted with data from certain population groups (e.g., students) or in particular areas and did not compare different settings [16,74]. We also investigated a large array of mediators, which have not been considered in previous studies [43]. Most associations between childhood NOE exposure and possible mediators pointed in a similar direction at pooled level and at the city-specific level, which suggests that these associations were independent of the various cultural contexts and living situations. The consistency of the associations between childhood NOE exposure and mental health and vitality, even after adjustment for potential mediators, shows the robustness of these relationships. 

### 4.4. Implications for Policymakers

Our study shows the importance of exposure to NOE during childhood for the development of a nature-appreciating attitude and healthy psychological state in adulthood. Currently, 73% of the population in Europe lives in urban areas [75]. Given that this number is projected to increase to be over 80% by 2050, it is important to recognize the implications for children growing up in environments with limited opportunities for exposure to NOE [75]. As an indoor lifestyle is highly prevalent among European children, it would be desirable to make NOE available, safe, and inviting places for children to play. In most countries, activities in nature are not a regular part of the school curriculum. Consequently, children who do not have opportunities to interact with and gain an appreciation of nature at home, have little chance to experience contact with nature. Our study showed that low levels of childhood NOE exposure are associated with worse mental health in adulthood, which supports the call on policymakers to improve availability of NOE for children. 

### 4.5. Future Research

Longitudinal studies that objectively measure childhood NOE exposure and health data are needed in order to investigate associations between accessibility of NOE, time and activities spent in nature during childhood, and mental as well as physical health throughout the life course. It is also unknown how childhood NOE exposure influences brain development in the long-term, and which mental health domains benefit from nature experience during childhood. Future investigations should also focus on how people actively search for contact with nature geographically further away, if NOE in the accessible living environment are scarce. 

## 5. Conclusions

This study suggests that childhood NOE exposure might be associated with mental well-being in adulthood. Low levels of childhood NOE exposure were associated with worse mental health, higher perceived amount of green and blue spaces, and with lower importance of NOE in adulthood. However, the association between childhood NOE exposure and mental health was not mediated by the perceived amount, use, satisfaction, importance or availability of NOE in adulthood. Further studies are needed to confirm these findings and to identify mechanisms underlying long-term benefits of childhood NOE exposure.

## Figures and Tables

**Table 1 ijerph-16-01809-t001:** Demographics of the study population: pooled and city-specific analyses.

Demographics	All Cities Pooled	Barcelona (Spain)	Doetinchem (The Netherlands)	Kaunas (Lithuania)	Stoke-on-Trent (UK)
*n* (%) total number of participants	3585	983 (27.4)	850 (23.7)	892 (24.9)	860 (24.0)
Mean (SD) age	51.7 (15.8)	45.2 (15.6)	56.4 (12.1)	60.0 (13.6)	46.0 (16.0)
*n* (%) gender					
Male	1616 (45.1)	466 (47.4)	372 (43.8)	360 (40.4)	418 (48.6)
Female	1969 (54.9)	517 (52.6)	478 (56.2)	532 (56.6)	442 (51.4)
*n* (%) born in country of residence					
“No”	350 (9.8)	235 (23.9)	34 (4.0)	35 (3.9)	46 (5.4)
“Yes”	3235 (90.2)	748 (76.1)	816 (96.0)	857 (96.1)	814 (94.7)
*n* (%) smoking status					
Current	730 (20.4)	269 (27.4)	121 (14.2)	101 (11.3)	239 (27.8)
Former	1009 (28.2)	227 (23.1)	377 (44.4)	232 (26.0)	173 (20.1)
Never	1846 (51.5)	487 (49.5)	352 (41.4)	559 (62.7)	448 (52.1)
*n* (%) highest education					
Low	252 (7.0)	146 (14.9)	10 (1.2)	16 (1.8)	80 (9.3)
Medium	1568 (43.7)	378 (38.5)	399 (46.9)	238 (26.7)	553 (64.3)
High	1765 (49.2)	459 (46.7)	441 (51.9)	638 (71.5)	227 (26.4)
*n* (%) perceived financial situation					
“Cannot make ends meet”	383 (10.7)	125 (12.7)	146 (17.2)	45 (5.0)	67 (7.8)
“Just have enough to get along”	1799 (50.2)	487 (49.5)	260 (30.6)	639 (71.6)	413 (48.0)
“Comfortable”	1403 (39.1)	371 (37.7)	444 (52.2)	208 (23.3)	380 (44.2)

*n*, number of participants; SD, standard deviation; UK, United Kingdom.

**Table 2 ijerph-16-01809-t002:** Current living arrangements of the study population: pooled and city-specific analyses.

Current Living Arrangements	All Cities Pooled	Barcelona (Spain)	Doetinchem (The Netherlands)	Kaunas (Lithuania)	Stoke-on-Trent (UK)
*n* (%) household composition					
“Living alone”	610 (17.0)	67 (6.8)	195 (22.9)	156 (17.5)	192 (22.3)
“Living with partner without children”	1240 (34.6)	242 (24.6)	387 (45.5)	353 (39.6)	258 (30.0)
“Living alone or with partner and children under 12 years”	534 (14.9)	184 (18.7)	126 (14.8)	38 (4.3)	186 (21.6)
“Living alone or with partner and only children ≥12 years”	602 (16.8)	143 (14.6)	129 (15.2)	191 (21.4)	139 (16.2)
“Other”	599 (16.7)	347 (35.3)	13 (1.5)	154 (17.3)	85 (9.9)
*n* (%) neighborhood socio-economic status					
Low	1127 (31.4)	325 (33.1)	266 (31.3)	227 (25.5)	309 (35.9)
Medium	1377 (38.4)	333 (33.9)	333 (39.2)	427 (47.9)	284 (33.0)
High	1081 (30.2)	325 (33.1)	251 (29.5)	238 (26.7)	267 (31.1)

*n*, number of participants; UK, United Kingdom.

**Table 3 ijerph-16-01809-t003:** NOE exposure, mental health and vitality of the study population: pooled and city-specific analyses.

NOE Exposure, Mental Health and Vitality	All Cities Pooled	Barcelona (Spain)	Doetinchem (The Netherlands)	Kaunas (Lithuania)	Stoke-on-Trent (UK)
*n* (%) childhood NOE exposure					
Low levels	562 (15.7)	184 (18.7)	60 (7.1)	100 (11.2)	218 (25.4)
High levels	3023 (84.3)	799 (81.3)	790 (92.9)	792 (88.8)	642 (74.7)
Median (IQR) perception of amount of NOE (scale 0–12)	7 (5)	5 (6)	9 (3)	8 (3)	7 (5)
Median (IQR) use of NOE (0–160 h per month)	4 (11.75)	4 (12)	10 (8.25)	4 (11.25)	1.25 (10)
Median (IQR) satisfaction with NOE (sum score scale 4–20)	15 (4)	14 (5)	16 (3)	14 (4)	16 (4)
Median (IQR) importance of NOE (sum score scale 4–20)	16 (4)	17 (3)	17 (3)	16 (3)	16 (3)
Median (IQR) residential surrounding greenness in 300 m (scale 0–1)	0.49 (0.24)	0.19 (0.09)	0.54 (0.12)	0.54 (0.10)	0.47 (0.10)
Median (IQR) transformed mental health (sum score scale 0–100)	76 (20)	72 (24)	84 (12)	72 (24)	76 (20)
Median (IQR) transformed vitality (sum score scale 0–100)	65 (25)	60 (25)	70 (20)	60 (25)	55 (25)

IQR, interquartile range; *n*, number of participants; NOE, natural outdoor environments; UK, United Kingdom.

**Table 4 ijerph-16-01809-t004:** Demographics of the study population on pooled level for subgroups with different amount of childhood NOE exposure (low levels and high levels).

Demographics	Low Levels Of Childhood NOE Exposure	High Levels Of Childhood NOE Exposure	*p*-Value *
*n* (%) total number of participants	562 (15.7)	3023 (84.3)	
Mean (SD) age	51.4 (15.5)	51.8 (15.9)	0.608
*n* (%) gender			0.666
Male	258 (45.9)	1358 (44.9)	
Female	304 (54.1)	1665 (55.1)	
*n* (%) born in country of residence			0.014
“No”	39 (6.9)	311 (10.3)	
“Yes”	523 (93.1)	2712 (89.7)	
*n* (%) smoking status			0.914
Current	112 (19.9)	618 (20.4)	
Former	162 (28.8)	847 (28.0)	
Never	288 (51.3)	1558 (51.5)	
*n* (%) highest education			<0.001
Low	56 (10.0)	196 (6.5)	
Medium	269 (47.9)	1299 (43.0)	
High	237 (42.2)	1528 (50.6)	
*n* (%) perceived financial situation			0.147
“Cannot make ends meet”	47 (8.4)	336 (11.1)	
“Just have enough to get along”	292 (52.0)	1507 (49.9)	
“Comfortable”	223 (39.7)	1180 (39.0)	

*n*, number of participants; SD, standard deviation. * based on *t*-tests, chi-square, and rank-sum tests and *p*-values statistically significant at 5% level while adjusted for the false discovery rate with the Benjamini–Hochberg Procedure.

**Table 5 ijerph-16-01809-t005:** Current living arrangements of the study population on pooled level for subgroups with different amount of childhood NOE exposure (low levels and high levels).

Current Living Arrangements	Low Levels Of Childhood NOE Exposure	High Levels Of Childhood NOE Exposure	*p*-Value *
*n* (%) household composition			0.616
“Living alone”	91 (16.2)	519 (17.2)	
“Living with partner without children”	204 (36.3)	1036 (34.3)	
“Living alone or with partner and children under 12 years”	82 (14.6)	452 (15.0)	
“Living alone or with partner and only children ≥12 years”	101 (18.0)	501 (16.6)	
“Other”	84 (15.0)	515 (17.0)	
*n* (%) neighborhood socio-economic status			0.979
Low	176 (31.3)	951 (31.5)	
Medium	218 (38.8)	1159 (38.3)	
High	168 (29.9)	913 (30.2)	

*n*, number of participants; * based on *t*-tests, chi-square, and rank-sum tests and *p*-values statistically significant at 5% level while adjusted for the false discovery rate with the Benjamini–Hochberg Procedure.

**Table 6 ijerph-16-01809-t006:** NOE exposure, mental health and vitality of the study population on pooled level for subgroups with different amount of childhood NOE exposure (low levels and high levels).

NOE Exposure, Mental Health and Vitality	Low Levels Of Childhood NOE Exposure	High Levels Of Childhood NOE Exposure	*p*-Value *
median (IQR) *n* perception of amount of NOE (scale 0–12)	7 (5) 560	7 (5) 3020	0.011
median (IQR) *n* use of NOE (scale 0–160 h)	3.75 (12) 553	4 (11.75) 2932	0.001
NOE median (IQR) *n* satisfaction with (scale 4-20)	15 (4) 560	15 (4) 3016	0.533
median (IQR) n importance of NOE (scale 4–20)	16 (3.5) 560	16 (4) 3020	0.250
median (IQR) n residential surrounding greenness in 300 m (scale 0–1)	0.45 (0.27) 562	0.49 (0.23) 3023	<0.001
median (IQR) n mental health (scale 0–100)	72 (20) 562	76 (24) 3021	<0.001
median (IQR) n vitality (scale 0–100)	60 (20) 562	65 (25) 3021	<0.001

IQR, interquartile range; *n*, number of participants; NOE, natural outdoor environments. * based on *t*-tests, chi-square, and rank-sum tests and *p*-values statistically significant at 5% level while adjusted for the false discovery rate with the Benjamini–Hochberg Procedure.

**Table 7 ijerph-16-01809-t007:** Mixed effects regression of childhood NOE exposure (low levels vs. high levels) on mental health and vitality (0–100 scales; higher is better mental health/vitality). Pooled and city-specific analyses.

Outcome Variables	*n* ^c^	Crude Model ^a^ β-Coefficient (95% CI)	Adjusted Main Model ^b^ β-Coefficient (95% CI)
Mental health			
Pooled	3583	−4.01 (−5.44, −2.59) ***	−4.13 (−5.52, −2.74) ***
Barcelona	983	−4.11 (−6.58, −1.64) **	−3.90 (−6.31, −1.50) **
Doetinchem	848	−1.07 (−4.65, 2.51)	−0.81 (−4.28, 2.66)
Kaunas	892	−4.16 (−7.69, −0.63) *	−3.58 (−7.01, −0.16) *
Stoke-on-Trent	860	−4.66 (−7.09, −2.22) ***	−5.48 (−7.87, −3.08) ***
Vitality			
Pooled	3583	−1.38 (−3.03, 0.28)	−1.29 (−2.90, 0.32)
Barcelona	981	−2.32 (−5.15, 0.51)	−1.97 (−4.75, 0.81)
Doetinchem	850	−0.88 (−5.39, 3.64)	−0.39 (−4.79, 4.00)
Kaunas	892	−4.17 (−7.70, −0.64) *	−3.73 (−7.14, −0.32) *
Stoke-on-Trent	860	1.16 (−1.92, 4.23)	0.21 (−2.80, 3.21)

* *p* ≤ 0.05; ** *p* < 0.01; *** *p* < 0.001. ^a^ Unadjusted model with random intercepts. ^b^ Adjusted for: age, gender, origin, education, perceived financial situation, socioeconomic status (SES) of the neighborhood, household composition, smoking status. ^c^
*n* was the same for the crude and the adjusted main model. CI, confidence interval; n, number of participants.

**Table 8 ijerph-16-01809-t008:** Mixed effects regression of childhood NOE exposure (low levels vs. high levels) on the amount of NOE, use of NOE, satisfaction with NOE, importance of NOE, and residential surrounding greenness in 300 m. Pooled and city-specific analyses.

Outcome Variables	*n* ^c^	Crude Model ^a^ Odds Ratio (95% CI)	Adjusted Model ^b^ Odds Ratio (95% CI)
Perception of amount of NOE			
Pooled	3580	1.34 (1.08, 1.66) **	1.32 (1.06, 1.64) *
Barcelona	982	0.69 (0.46, 1.04)	0.72 (0.47, 1.10)
Doetinchem	850	0.52 (0.29, 0.91) *	0.45 (0.25, 0.81) **
Kaunas	892	0.95 (0.61, 1.48)	0.96 (0.61, 1.50)
Stoke-on-Trent	856	1.69 (1.20, 2.38) **	1.61 (1.13, 2.30 ) **
Use of NOE			
Pooled	3485	0.92 (0.75, 1.12)	0.92 (0.75, 1.12)
Barcelona	970	0.95 (0.67, 1.35)	0.94 (0.65, 1.36)
Doetinchem	842	1.47 (0.84, 2.54)	1.49 (0.84, 2.65)
Kaunas	817	0.66 (0.42, 1.02)	0.59 (0.37, 0.93) *
Stoke-on-Trent	856	0.75 (0.54, 1.05)	0.78 (0.56, 1.11)
Satisfaction with NOE			
Pooled	3576	1.06 (0.83, 1.24)	1.02 (0.83, 1.25)
Barcelona	977	0.61 (0.43, 0.88) **	0.64 (0.44, 0.94) *
Doetinchem	850	0.70 (0.37, 1.30)	0.67 (0.35, 1.26)
Kaunas	892	1.34 (0.88, 2.03)	1.34 (0.87, 2.05)
Stoke-on-Trent	857	1.21 (0.84, 1.74)	1.17 (0.80, 1.69)
Importance of NOE			
Pooled	3580	0.79 (0.66, 0.96) *	0.81 (0.66, 0.98) *
Barcelona	981	0.79 (0.56, 1.11)	0.77 (0.54, 1.10)
Doetinchem	849	1.05 (0.61, 1.83)	1.02 (0.57, 1.81)
Kaunas	892	0.69 (0.43, 1.10)	0.77 (0.47, 1.24)
Stoke-on-Trent	858	0.83 (0.60, 1.16)	0.83 (0.59, 1.16)
Residential surrounding greenness in 300 m			
Pooled	3585	0.79 (0.57, 1.08)	0.77 (0.56, 1.05)
Barcelona	983	0.77 (0.34, 1.74)	0.83 (0.35, 1.97)
Doetinchem	850	0.74 (0.36, 1.51)	0.69 (0.33, 1.46)
Kaunas	892	0.88 (0.56, 1.38)	0.79 (0.50, 1.27)
Stoke-on-Trent	860	1.03 (0.65, 1.64)	0.94 (0.58, 1.52)

* *p* ≤ 0.05; ** *p* < 0.01. ^a^ Unadjusted model with random intercepts. ^b^ Adjusted for: age, gender, origin, education, perceived financial situation, SES of the neighborhood, household composition, smoking status. ^c^
*n* was the same for the crude and the adjusted model. CI, confidence interval; *n*, number of participants; NOE, natural outdoor environments.

**Table 9 ijerph-16-01809-t009:** Mixed effects regression of childhood NOE exposure (low levels vs. high levels) on mental health and vitality (0–100 scales; higher scores indicate better mental health/vitality) with additional adjustments for the perception of amount of NOE, use of NOE, and satisfaction with NOE. Pooled and city-specific analyses.

Outcome Variables	Additional Adjustment for Perception of Amount of NOE β-Coefficient (95% CI), *n*	Additional Adjustment for Use of NOE β-Coefficient (95% CI), *n*	Additional Adjustment for Satisfaction with NOE β-Coefficient (95% CI), *n*
Mental health			
Pooled	−4.21 (−5.97, −2.82) ***, 3578	−4.16 (−5.55, −2.77) ***, 3483	−4.20 (−5.58, −2.81) ***, 3574
Barcelona	−3.81 (−6.22, −1.40) **, 982	−3.84 (−6.27, −1.42) **, 970	−3.64 (−6.03, −1.24) **, 977
Doetinchem	−0.70 (−4.17, 2.78), 848	−0.89 (−4.35, 2.57), 840	−0.82 (−4.25, 2.61), 848
Kaunas	−3.68 (−7.09, −0.27) *, 892	−3.77 (−7.26, −0.29) *, 817	−3.78 (−7.21, −0.36) *, 892
Stoke-on-Trent	−5.51 (−7.93, −3.08) ***, 856	−5.54 (−7.93, −3.14) ***, 856	−5.63 (−8.03, −3.23) ***, 857
Vitality			
Pooled	−1.39 (−3.00, 0.22), 3578	−1.26 (−2.88, 0.36), 3483	−1.34 (−2.95, 0.26), 3574
Barcelona	−1.85 (−4.63, 0.93), 980	−1.83 (−4.63, 0.98), 968	−1.60 (−4.38, 1.18), 975
Doetinchem	−0.16 (−4.54, 4.23), 850	−0.52 (−4.90, 3.86), 842	−0.42 (−4.77, 3.94), 850
Kaunas	−3.82 (−7.22, −0.41) *, 892	−3.54 (−7.04, −0.04) *, 817	−3.84 (−7.26, −0.43) *, 892
Stoke-on-Trent	−0.10 (−3.13, 2.93), 856	0.02 (−2.97, 3.01), 856	−0.20 (−3.17, 2.77), 857

* *p* ≤ 0.05; ** *p* < 0.01; *** *p* < 0.001. CI, confidence interval; *n*, number of participants; NOE, natural outdoor environments.

**Table 10 ijerph-16-01809-t010:** Mixed effects regression of childhood NOE exposure (low levels vs. high levels) on mental health and vitality (0–100 scales; higher is better mental health/vitality) with additional adjustments for importance of NOE, residential surrounding greenness within 300 m, and all mediators combined. Pooled and city-specific analyses.

Outcome Variables	Additional Adjustment for Importance of NOE β-Coefficient (95% CI), *n*	Additional Adjustment for Residential Surrounding Greenness within 300 mβ-Coefficient (95% CI), *n*	Additional Adjustment for all Mediators Combined ^a^ β-Coefficient (95% CI), *n*
Mental health			
Pooled	−4.15 (−5.54, −2.76) ***, 3578	−4.12 (−5.50, −2.72) ***, 3583	−4.28 (−5.67, −2.89) ***, 3464
Barcelona	−3.93 (−6.35, −1.52) **, 981	−3.90 (−6.31, −1.49) **, 983	−3.70 (−6.13, −1.27) **, 961
Doetinchem	−0.80 (−4.27, 2.67), 847	−0.80 (−4.27, 2.67), 848	−1.00 (−4.43, 2.43), 839
Kaunas	−3.57 (−6.99, −0.14) *, 892	−3.62 (−7.05, −0.20) *, 892	−4.11 (−7.59, −0.64) *, 817
Stoke-on-Trent	−5.48 (−7.87, −3.09) ***, 858	−5.48 (−7.88, −3.08) ***, 860	−5.73 (−8.15, −3.31) ***, 847
Vitality			
Pooled	−1.24 (−2.85, 0.36), 3578	−1.26 (−2.87, 0.35), 3483	−1.29 (−2.91, 0.33), 3464
Barcelona	−1.87 (−4.65, 0.91), 979	−1.95 (−4.72, 0.83), 981	−1.49 (−4.30, 1.33), 959
Doetinchem	−0.37 (−4.76, 4.02), 849	−0.38 (−4.77, 4.00), 850	−0.45 (−4.81, 3.92), 841
Kaunas	−3.71 (−7.12, −0.30) *, 892	−3.79 (−7.21, −0.38) *, 892	−3.76 (−7.25, −0.26) *, 817
Stoke-on-Trent	0.34 (−2.64, 3.32), 858	0.22 (−2.79, 3.23), 860	−0.31 (−3.29, 2.66), 847

* *p* ≤ 0.05; ** *p* < 0.01; *** *p* < 0.001. ^a^ “All mediators” includes: perception of amount of NOE, use of NOE, satisfaction with NOE, importance of NOE, NDIV within 300 m. CI, confidence interval; *n*, number of participants; NOE, natural outdoor environments.

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
