# Peer review of "Low Childhood Nature Exposure is Associated with Worse Mental Health in Adulthood"

_ijerph, 2019, doi:10.3390/ijerph16101809_

Round 1
Reviewer 1 Report
Review of journal article:
‘Low Childhood Nature Exposure is Associated with Worse Mental Health in Adulthood’
For The Journal of Environmental Research and Public Health
This manuscript looked at the level of childhood exposure to nature and the effects on mental health is adulthood. They had a large sample size (3585 participants) from four European cities. It was a questionnaire study. Adults with less outdoor exposure in childhood had significant worse Mental Health (MH) as adults. Children with less nature exposure as children perceived natural exposure as less important in adulthood. Limitations include the study being a cross sectional questionnaire study. Could other variables have been controlled for such as parental mental health issues? Retrospective reporting of childhood exposure to nature may also be an issue? Vitality seemed like an add on variable and I could not see clearly the argument for why it was included? This study does tackle an important and topical area of study. Please see my comments in the specific manuscript sections below on how to improve on this.
Title:
The title is fine as it describes the main finding of the study.
Abstract:
The abstract is mostly well written and structured. Please find below some suggestions to improve it.
1. The first sentence is missing a word to make it grammatical. It should be “Nature exposure is associated with favourable brain development and physical and mental health’.
2. The abstract may already be to the word count so may not be able to be expanded upon. However, it is good to see abstracts that also point out the main limitations of the study so that results are interpreted with caution.
3. In the Methods section, it is not clear who completed the questionnaires? The participants as adults or their parents? The design of the study so far from just reading the title and abstracts raises questions as to who would have been the best people to complete the questionnaires on the participants’ exposure to nature as children? Would the adult participants and their parents have been the best and most reliable sources? That way you could have looked to see if there was any discrepancies in the responses from the parents and their children. Children have infantile amnesia when younger and would not be able to recall where they lived under the age of around 2 years old or how much exposure they had to nature etc?
4. Mental health and vitality are two interesting concepts to study together? Vitality is a less used term and one that the readers may be less familiar with than mental health. Do you explain what you mean by vitality?
5. The authors also mention a result about the perceived importance of childhood Natural Outdoor Environment (NOE) exposure however, it was not mentioned as one of the aims of the study?
Introduction:
The introduction was ok. It was a little short and I would have liked to see a more biopsychosocial model of childhood NOE exposure discussed. See my points below on how to improve on this:
1. Page 2, line 54 and 55, remove mention of your current study here as it is confusing. Just mention your study at the end of the literature review. This section is just for a review of the existing relevant literature and to build an argument for the need of your study.
2. You need to define the acronym NOE when you first use it in the literature review/introduction which appears to be page 2, line 56 however, it is not defined until page 2, line 67?
3. There is numerous mention of how NOE can help with mental health and brain development. Can you please supple a few sentences on the biological effects of NOE in childhood that would lead to this finding? Is it related to brain chemistry such as dopamine etc?? A biopsychosocial model of the benefits of exposure to NOE in childhood could be discussed in that order? At present the structure of writing of the benefits of NOE childhood exposure is a bit less unclear.
4. Define BMI acronym when you first use it (page 2, line 69).
5. You need to define the term vitality? It is not as commonly researched as MH therefore, why it is included in the study is unclear? You need at least a paragraph on vitality, define the term and what is known about it and NOE exposure in childhood etc.
6. Also the issue around childhood exposure to NOE and the effects on adults’ perceived importance of NOE needs a paragraph on its own as well.
7. Vitality and perceived importance of childhood NOE are not discussed in the literature review/introduction just in one line towards the end of this section?
8. You need a summary paragraph of the main findings of the literature review/introduction.
Methods
Again this was an Ok section. See points for improving on this below.
1. Start with the ethics information and then move onto the participants’ information.
2. Within the final sample, were there a relative even number of participant recruited from each of the four European cities? Or did some cities have more participants with missing data? Could you have a flow chart figure here re: participants included and not included and why?
3. Was the age the participant was at the time completing the questionnaire associated with missing data on their childhood information (such as the longer the time to have to recall this information an issue)?
4. The terms on the Likert scale for NOE exposure in childhood sounded a bit vague. What did “sometimes mean” such as once a week/month etc? I know it has come from a previous study but could it have been improved for the current one?
5. Having parents or older siblings (where possible) etc of the participants also complete some of the questionnaire for the participant (questions on their childhood) may help verify the participants’ data?
6. Did NOE exposure include participants’ day at their school or where you asking for their time with family and friends outside of school or both? Can you please be a bit more clear on the methodology of this part of the study given it the man part?
7. Did you ask if people had lived in high rise apartments versus houses with gardens?
8. Did you ask about the participants’ family history of mental health issues? Could you have controlled for this in your study?
9. Did you ask if they went on holidays as children and where too? Especially as you included questions about exposure to water.
10. Did you ask the SES of the family when the participants was a child? Did you ask if they lived in high density living?
11. Did you ask if they lived in private or public accommodation as children?
12. Did you ask about their school’s physical environment given the amount of time they spent there?
13. I suspect some of the cities exposure to the outdoors is quite seasonal, such as more time spent outside in the warmer months? Did you take that into account in your questionnaire?
14. Do some of the cities you recruited from have more outdoor areas for people to use for free?
Results
1. The majority of the sample had high level of Childhood NOE exposure (84.3% compared to 15.7% with low levels). Was this expected? I did not see this addressed in the discussion?
2. The results tables on participants are very long. Could you split them up a bit better so there are more but with less variables, and include related variables to each table more such as demographics in one table (age, gender, born in country of residence etc), then current living arrangements as another table, then the NOE childhood exposure etc as another table? Try and avoid having tables that go over a page.
3. Same for Table 2 to be broken down into smaller tables as suggested above?
4. The analyses appeared ok, however the verdict from a statistician may be good as it is a big data set with many variables. There were a number of variables that could have been controlled for.
Discussion:
It was a good start to the discussion reminding the reader what the study was about. I like that subheadings were used for structure.
1. The discussion starts talking about potentially underlying mechanisms? I think this is what is suited more to the introduction. Try and not to introduce any new information here besides talking about your results and relating it to past research.
2. The discussion could go through each main result and explain relates to or doesn’t relate to the existing research.
3. A limitation you have left out is not asking the participants for their family history of mental health issues and controlling for this in your statistical analyses. Other limitations have been discussed above
References
There were a good list of references.
1. Some minor formatting errors were noticed such as inconsistency with using capitals in journal names.
Author Response
Reviewer 1
Review of journal article:
‘Low Childhood Nature Exposure is Associated with Worse Mental Health in Adulthood’
For The Journal of Environmental Research and Public Health
This manuscript looked at the level of childhood exposure to nature and the effects on mental health is adulthood. They had a large sample size (3585 participants) from four European cities. It was a questionnaire study. Adults with less outdoor exposure in childhood had significant worse Mental Health (MH) as adults. Children with less nature exposure as children perceived natural exposure as less important in adulthood. Limitations include the study being a cross sectional questionnaire study. Could other variables have been controlled for such as parental mental health issues? Retrospective reporting of childhood exposure to nature may also be an issue? Vitality seemed like an add on variable and I could not see clearly the argument for why it was included? This study does tackle an important and topical area of study. Please see my comments in the specific manuscript sections below on how to improve on this.
Title:
The title is fine as it describes the main finding of the study.
Abstract:
The abstract is mostly well written and structured. Please find below some suggestions to improve it.
1. The first sentence is missing a word to make it grammatical. It should be “Nature exposure is associated with favourable brain development and physical and mental health’.
Authors: Thank you for the notification. The sentence was corrected and changed to:
“Exposure to natural outdoor environments (NOE) is associated with health benefits, however, evidence on the impact of NOE exposure during childhood on mental health (MH) and vitality in adulthood is scarce.
2. The abstract may already be to the word count so may not be able to be expanded upon. However, it is good to see abstracts that also point out the main limitations of the study so that results are interpreted with caution.
Authors: We thank the reviewer for this suggestion but we are afraid that the word limit does not allow for further expansion of the abstract.
3. In the Methods section, it is not clear who completed the questionnaires? The participants as adults or their parents? The design of the study so far from just reading the title and abstracts raises questions as to who would have been the best people to complete the questionnaires on the participants’ exposure to nature as children? Would the adult participants and their parents have been the best and most reliable sources? That way you could have looked to see if there was any discrepancies in the responses from the parents and their children. Children have infantile amnesia when younger and would not be able to recall where they lived under the age of around 2 years old or how much exposure they had to nature etc?
Authors:
In the abstract the sentence: “This study was based on questionnaire data from 3585 participants, aged 18-75, collected in the PHENOTYPE project…” has been changed to “This study was based on questionnaire data collected from 3585 participants, aged 18-75, in the PHENOTYPE project…” in order to clarify that the data was collected from the participants, aged 18-75, themselves.
For further clarification, we have also changed the sentence in the Methods section (p. 4, lines 163-166) “Childhood NOE exposure was assessed by asking participants to rate how often they visited NOE during their childhood.” to “Participants were asked to retrospectively rate how often they spent time in NOE during their childhood.”
We agree that retrospective data collection contains a potential for bias. It is for this reason that the section “Strengths and limitations” contains that: “Our cross-sectional study, by design, had a limited capability to establish a causal relationship. Moreover, our exposure variable (childhood nature exposure) was assessed with only one question and, as it was measured retrospectively, there is the potential for recall bias”. This section was now complemented by adding (p. 23, lines 484-486): “A longitudinal study design or the additional assessment of childhood NOE by the participants’ parents would have increased the reliability of data on childhood NOE.”
4. Mental health and vitality are two interesting concepts to study together? Vitality is a less used term and one that the readers may be less familiar with than mental health. Do you explain what you mean by vitality?
Authors: Thank you for the notification. We have included a short description in the abstract: “Mixed models were used to investigate associations between childhood NOE exposure and (i) MH; (ii) vitality (perceived level of energy and fatigue); and (iii) potential mediation by perceived amount, use, satisfaction, importance of NOE, and residential surrounding greenness, using pooled and city-level data.”
A detailled description of the variable vitality is provided in the methods section (p. 7, lines 207-209) and has been complemented to: “The 4-item vitality part concerned perceived level of energy and fatigue and is recommended in order to better capture differences in subjective well-being”.
5. The authors also mention a result about the perceived importance of childhood Natural Outdoor Environment (NOE) exposure however, it was not mentioned as one of the aims of the study?
Authors:
We have clarified this in the abstract by naming all the mediators that are being investigated. It is now: “Mixed models were used to investigate associations between childhood NOE exposure and (i) MH; (ii) vitality (perceived level of energy and fatigue); and (iii) potential mediation by perceived amount, use, satisfaction, importance of NOE, and residential surrounding greenness, using pooled and city-level data.”.
Introduction:
The introduction was ok. It was a little short and I would have liked to see a more biopsychosocial model of childhood NOE exposure discussed. See my points below on how to improve on this:
1. Page 2, line 54 and 55, remove mention of your current study here as it is confusing. Just mention your study at the end of the literature review. This section is just for a review of the existing relevant literature and to build an argument for the need of your study.
Authors: Thank you, we have removed the part of the sentence as you suggested and it is now: “Previous studies found that green spaces positively influence physical and mental health as well as general well-being in adults”.
2. You need to define the acronym NOE when you first use it in the literature review/introduction which appears to be page 2, line 56 however, it is not defined until page 2, line 67?
Authors: Thank you. NOE is now defined where it first appears, which is in the following sentence:
“In contrast, the availability of natural outdoor environments (NOE) and the time that children spend in them have been associated with increased self-esteem, quality of life, respiratory health, physical activity, and lower body mass index (BMI) [31-34].” (line 70-73)
3. There is numerous mention of how NOE can help with mental health and brain development. Can you please supple a few sentences on the biological effects of NOE in childhood that would lead to this finding? Is it related to brain chemistry such as dopamine etc?? A biopsychosocial model of the benefits of exposure to NOE in childhood could be discussed in that order? At present the structure of writing of the benefits of NOE childhood exposure is a bit less unclear.
Author: Thank you for this suggestion. Changes have been made in the introduction line 73-78:
“It has also been suggested that NOE provide benefits that directly influence cognitive development in children, as NOE provide opportunities that stimulate discovery, creativity, mastery, engagement, risk taking behavior, engagement, basic emotional states (e.g. surprise), and psychological restoration [35-37]. Ways in which NOE are suggested to exert indirect beneficial effects on the cognitive development of children include the mitigation of traffic-related air pollution, reduction of noise, and increased levels of physical activity [11,35,38-41].”
4. Define BMI acronym when you first use it (page 2, line 69).
Authors: This has been corrected and is now: “In contrast, the availability of NOE and the time that children spend in them have been associated with increased self-esteem, quality of life, respiratory health, physical activity, and lower body mass index (BMI) [31-34].”
5. You need to define the term vitality? It is not as commonly researched as MH therefore, why it is included in the study is unclear? You need at least a paragraph on vitality, define the term and what is known about it and NOE exposure in childhood etc.
Authors: Short information on vitality is now provided in the abstract as mentioned above. We have added information explaining why we have included vitality in the introduction (p. 2, lines 84-88): “Previous studies also reported on the association between time spent in green spaces in adulthood and higher levels of vitality [44]. Associations have also been reported for the amount of green spaces in the neighborhood and vitality, perceived general health, and greater quality of life in adults [45-47]. To our knowledge, however, the association between childhood NOE exposure and vitality in adulthood has not been analyzed yet.”
6. Also the issue around childhood exposure to NOE and the effects on adults’ perceived importance of NOE needs a paragraph on its own as well.
Authors: We have accentuated that paragraph in the introduction by adding an introductory sentence to the section (p. 3, lines 89-91):
“Furthermore, some authors have suggested that childhood NOE exposure can have long-term influence on perceptions, evaluations and use of NOE, but evidence on this proposed relationship is scarce.”
Following that sentence, previous literature that concerns the association between childhood NOE experience and attitude towards nature, approach of green spaces, perception of natural environments, and interaction with nature is discussed. We finish that paragraph now with (p. 3, lines 99-106):
“These findings show that childhood NOE exposure could potentially influence the way people interact with NOE in adulthood. It might also affect the choice for residential surrounding greenness and the perceived amount, use, satisfaction, and importance of NOE in adulthood. These factors could subsequently lead to health benefits and mediate the association between childhood NOE exposure and mental health or vitality. Therefore, it is important to consider these variables as potential mediators in the analysis.”
7. Vitality and perceived importance of childhood NOE are not discussed in the literature review/introduction just in one line towards the end of this section?
Authors: We have added information on vitality (as described above under point 5) and about the mediators (as described under point 6) in the introduction.
8. You need a summary paragraph of the main findings of the literature review/introduction.
Authors: Thank you for this suggestion. A summary paragraph was added in line 103-105: “It can be concluded, that existing literature highlights the importance of childhood NOE exposure for physical, mental, and cognitive development. High levels of time spent in NOE at adulthood are associated with favorable mental health and vitality states at adulthood.”
Methods
Again this was an Ok section. See points for improving on this below.
1. Start with the ethics information and then move onto the participants’ information.
Authors: A separate sub-heading “Ethical approval” at the beginning of the methods section has been added. It also includes all ethical codes.
“Ethical approval
The appropriate ethical committees of the involved research institutes approved data collection in conjunction with the “Positive health effects of the natural outdoor environment in typical populations in different regions in Europe” (PHENOTYPE) project: Parc de Salut Mar, Clinical Research Ethics Committee reference number 2011/4206/I; Medisch Ethische Toetsingscommissie UMCU reference number 12/595; Lietuvos Bioetikos Komitetas reference number 6B-12-147; Faculty of Health Sciences’ Ethics Panel, no reference number). Written informed consent was obtained from all participants [44,52]. The study was conducted in accordance with the Declaration of Helsinki [53].“
2. Within the final sample, were there a relative even number of participant recruited from each of the four European cities? Or did some cities have more participants with missing data? Could you have a flow chart figure here re: participants included and not included and why?
Authors: Initially, 1000 participants were recruited per city. Data of 3986 participants was available and data of 3585 participants contained all necessary variables. These participants were equally distributed among the cities, with the lowest number of participants in Doetinchem (850, equal to 23.7% of the total study population) and the highest number in Barcelona (983, equal to 27.4% of the total study population). These numbers are also displayed in Table 1.1.
3. Was the age the participant was at the time completing the questionnaire associated with missing data on their childhood information (such as the longer the time to have to recall this information an issue)?
Authors: The variable “childhood NOE exposure” was only one variable that led to exclusion when it was missing. Other variables that led to exclusion when information was missing were: age, gender, education, perceived financial situation, SES of the neighborhood, household composition, and smoking status. Childhood NOE exposure was therefore only one out of eight variables that led to exclusion. The mean age of the sample that did not answer the question on childhood nature exposure (n=15) was 49.8 years, compared to a mean of 51.7 years of the sample that did answer the question. We therefore do not think that age at the time of questionnaire completion has affected the missingness of the variable childhood NOE exposure.
4. The terms on the Likert scale for NOE exposure in childhood sounded a bit vague. What did “sometimes mean” such as once a week/month etc? I know it has come from a previous study but could it have been improved for the current one?
Authors: The Likert scale was a 5-point scale with as answer options: never, sometimes, regularly, often, very often. We agree that the terms leave space for individual interpretation, but stricter defined answer options (e.g. every day, 2-3 times per week, 6-7 days per week) would have been really difficult to answer retrospectively and could therefore have increased recall bias.
5. Having parents or older siblings (where possible) etc of the participants also complete some of the questionnaire for the participant (questions on their childhood) may help verify the participants’ data?
Authors: This would have been an option to increase the reliability of data indeed. Data of family members, however, was not collected within the PHENOTYPE study. It should also be considered that the mean age of the participants was 51.7. Assuming an age of 27 for the first child would mean that the parents were around 80 years old at the time of data collection. At that age it might be cognitively hard to remember and fill in questionnaires/be interviewed and there is a realistic chance that some parents have died already. Still, we have incorporated this idea in the discussion under the section “strengths and limitations of the study” (please see line 477-479):
“A longitudinal study design or the additional assessment of childhood NOE by the participants’ parents or siblings would have increased the reliability of data on childhood NOE [42].”
6. Did NOE exposure include participants’ day at their school or where you asking for their time with family and friends outside of school or both? Can you please be a bit more clear on the methodology of this part of the study given it the man part?
Authors: The question about the time spent in NOE during childhood included all time that was spent in nature. This could be visits on purpose with the family (e.g. hiking in forests) or visits that are not primarily done on purpose (e.g. playing with friends in the backyard, climbing trees etc.). This information has also been added to line 159-162:
“Participants were asked to retrospectively rate how often they spent time in NOE during their childhood, including purposeful visits (e.g. hiking in natural parks with the parents) and non-purposeful visits (e.g. playing outside in the backyard).”
7. Did you ask if people had lived in high rise apartments versus houses with gardens?
Authors: As participants were only asked about their current living situation, we included this comment in the discussion as a limitation. Please see line 479-482:
“Additionally, information on the housing situation (high-rise vs. houses, with or without garden) and information on the family history of mental health issues would have provided a better picture of the childhood situation and it would have been favorable to control for these variables.”
8. Did you ask about the participants’ family history of mental health issues? Could you have controlled for this in your study?
Authors: The family history of physical or mental health issues were not part of the PHENOTYPE study and could therefore unfortunately not be taken into consideration. As a consequence, we added this in the discussion under “Strengths and limitations of the study” in line 479-482:
“Additionally, information on the housing situation (high-rise vs. houses, with or without garden) and information on the family history of mental health issues would have provided a better picture of the childhood situation and it would have been favorable to control for these variables.”
9. Did you ask if they went on holidays as children and where too? Especially as you included questions about exposure to water.
Authors: We did not have this information. It might not have affected the study, as tourism was not as prominent in the 1940’s-1980’s as it is now. This can also be seen in the following visualisation concerning worldwide tourism between 1950 and 2016, derived from ourworldindata.org/tourism.
Source: Max Roser (2019) - "Tourism". Published online at OurWorldInData.org. Retrieved from: 'https://ourworldindata.org/tourism' [Online Resource] (accessed on 2019-04-19)
10. Did you ask the SES of the family when the participants was a child? Did you ask if they lived in high density living?
Authors: To us, this question is similar to question 7 and the following question number 11. We can imagine that children who live in high rise, public accommodations in dense areas spent less time in NOE. In the end, however, this comes down to the essential question namely, how much time the participant actually spent in NOE during childhood. Still, we agree that it would have been interesting to see the results when controlled for SES and housing situation during childhood, but this was not possible with the available data.
11. Did you ask if they lived in private or public accommodation as children?
Authors: Please see our answer above. We included the limitation that there was no data on the housing situation during childhood available under the limitations of the study in line 479-480:
“Additionally, information on the housing situation (high-rise vs. houses, with or without garden) and information on the family history of mental health issues would have provided a better picture of the childhood situation and it would have been favorable to control for these variables.”
12. Did you ask about their school’s physical environment given the amount of time they spent there?
Authors: Descriptions on the physical environments of schools were not available and could therefore not be taken into consideration.
13. I suspect some of the cities exposure to the outdoors is quite seasonal, such as more time spent outside in the warmer months? Did you take that into account in your questionnaire?
Authors: The possible influence of climatic differences is discussed in the discussion in the section “strengths and limitations of the study”. Please see line 489-492:
“Also, we did not explicitly take micro- or macro-climatic differences and differences in vegetation types between the cities into account. These may have partly influenced the results, however, by using a mixed effects model with city as a random intercept, we have, to a large extent, accounted for these differences [69,70].”
14. Do some of the cities you recruited from have more outdoor areas for people to use for free?
Authors: Participants can have grown up anywhere, not necessarily in one of the four cities. Therefore, the amount of free NOE in the four cities is not influential on the variable childhood NOE exposure. For the variable residential surrounding greenness, the Normalized Difference Vegetation Index (NDVI) was used. The NDVI measures healthy vegetation, but does not provide information on which spaces are public or private.
Results
1. The majority of the sample had high level of Childhood NOE exposure (84.3% compared to 15.7% with low levels). Was this expected? I did not see this addressed in the discussion?
Authors: We also noticed this difference between the two groups. It is for this reason that we also conducted the sensitivity analyses for different cut offs (“never”, “sometimes”, “regularly” as low levels; “often”, “very often” as high levels). In the pooled analyses, low levels of childhood NOE exposure were still significantly associated with lower mental health scores, compared with high levels of childhood NOE exposure. The findings of the different cut offs are described under the heading “sensitivity analyses” in line 378-389. Still, future studies could measure childhood NOE exposure more extensive. We describe this in the Discussion (page 20):
“Moreover, our exposure variable (childhood nature exposure) was assessed with only one question and, as it was measured retrospectively, there is the potential for recall bias [67]. A longitudinal study design or the additional assessment of childhood NOE by the participants’ parents or siblings would have increased the reliability of data on childhood NOE [42].”
2. The results tables on participants are very long. Could you split them up a bit better so there are more but with less variables, and include related variables to each table more such as demographics in one table (age, gender, born in country of residence etc), then current living arrangements as another table, then the NOE childhood exposure etc as another table? Try and avoid having tables that go over a page.
Authors: Thank you for that idea. We broke up the table into 3 smaller tables (demographics, current living arrangements, and NOE exposure).
3. Same for Table 2 to be broken down into smaller tables as suggested above?
Authors: We also broke up the second table into 3 smaller tables.
4. The analyses appeared ok, however the verdict from a statistician may be good as it is a big data set with many variables. There were a number of variables that could have been controlled for.
Authors: We agree that there are variables that would have been informative, but unfortunately have not been collected and that we therefore could not control for.
Discussion:
It was a good start to the discussion reminding the reader what the study was about. I like that subheadings were used for structure.
1. The discussion starts talking about potentially underlying mechanisms? I think this is what is suited more to the introduction. Try and not to introduce any new information here besides talking about your results and relating it to past research.
Authors:
Information about cognitive development is now also mentioned in the introduction. We adapted the Discussion section so that it starts with the main findings and continues with the discussion of the association between childhood NOE exposure and mental health. We then continued discussing the association on childhood NOE exposure and vitality. For this, the following paragraph (line 438-446) has been added:
“Available data and information on the association between childhood NOE exposure and vitality is scarce. Our study revealed a non-significant association for low levels of childhood NOE exposure, when compared to high levels of childhood NOE exposure, and lower vitality scores. Van den Berg et al. found that childhood NOE exposure did not interact with the association between green space visits in adulthood and vitality in adulthood [44]. A possible explanation is that feelings of vitality are not as long-term influenced as mental health and rather depend on recently undertaken restoration and regeneration measures. Another possible explanation for this finding is that the SF-36 vitality scale comprises components that refer to physical health. Therefore, it might be stronger related to a currently active lifestyle than the SF-36 mental health scale [65].”
We further included a discussion on the results of the mediation analysis (lines 455-466):
“The mediation analysis revealed that low levels of childhood NOE exposure were associated with a higher perceived amount of NOE, when compared to high levels of childhood NOE exposure. This was contrary to the initial hypothesis. A potential explanation could be that that those with high levels of childhood NOE exposure have become less sensitive to the amount of NOE around them, as they had regular contact with NOE during their early life. This theory would also provide an explanation for the findings that participants with low levels of childhood NOE exposure rated the importance of NOE lower when compared to participants with high levels of childhood NOE exposure. Participants with low levels of childhood NOE exposure might be used to little contact with nature, while participants with high levels of childhood NOE exposure might take NOE for granted. Similar findings were also reported by van den Berg et al. who found that the mental health benefits gained from NOE visits in adulthood were stronger for people with low levels of childhood NOE exposure than for people with high levels of childhood NOE exposure [44].”
2. The discussion could go through each main result and explain relates to or doesn’t relate to the existing research.
Authors: Please see the answer above.
3. A limitation you have left out is not asking the participants for their family history of mental health issues and controlling for this in your statistical analyses. Other limitations have been discussed above
Authors: These suggestions have been added to the discussion in line 478-481.
“Additionally, information on the housing situation (high-rise vs. houses, with or without garden) and information on the family history of mental health issues would have provided a better picture of the childhood situation and it would have been favorable to control for these variables.”
References
There were a good list of references.
1. Some minor formatting errors were noticed such as inconsistency with using capitals in journal names.
Authors: The endnote reference style has been changed to MDPI and the inconsistency of one manually added reference has been corrected.
Reviewer 2 Report
In this work the authors determined whether childhood exposure to natural outdoor environment (NOE) had an impact on mental health and physical well-being in adulthood. This study was performed using questionnaire data that was collected from 3585 participants living in 4 European studies. Statistical analyses were performed to identify associations between childhood NOE exposure and (i) mental health scores; (ii) vitality (perceived level of energy & fatigue); and (iii) potential mediators (such as residential greenness), using pooled and city-level data. Results showed that low levels of childhood NOE exposure were associated with worse mental health in adulthood and not with vitality. No evidence of mediation was found. This article would be of interest to readers of IJERPH and is well written and researched. One drawback from the experimental design was that only 1 question was used to assess retrospective childhood NOE, but this fact was clearly outlined in the Discussion.
Major problems:
none
Minor Problems:
Data for the main finding in Table 3 concerning the association between mental health and childhood NOE exposure is consistent for pooled data and for the cities of Barcelona, Kaunas and Stoke-on-Trent, but there is a clear disconnect for Doetinchem which was not statistically significant. The authors should note this finding and postulate why the results for Doetinchem differ from the other cities.
By looking at current residential greenness, is the assumption made that childhood conditions were the same, or was the objective to determine whether adulthood residential greenness could alleviate a lack of childhood NOE? Thus, it is not entirely clear how adulthood residential greenness status could ‘mediate’ the association between childhood NOE and mental health.
Lines 28-29: first sentence of the Abstract is not a complete sentence.
Lines 52-53: change “Mental health is negatively influenced…” to “Mental health can be negatively influenced…” and “others exposure” to “other exposures”
Lines 54-55: delete “, in contrast to our study,”
Line 56: define NOE here rather than on line 67
Line 69: define BMI
Line 333: change ‘study’ to ‘studies’
Line 336: delete the first comma
Line 414: change “the nature” to “nature”
Line 420: change “with” to “which”
Line 423: delete the comma
Author Response
Reviewer 2
In this work the authors determined whether childhood exposure to natural outdoor environment (NOE) had an impact on mental health and physical well-being in adulthood. This study was performed using questionnaire data that was collected from 3585 participants living in 4 European studies. Statistical analyses were performed to identify associations between childhood NOE exposure and (i) mental health scores; (ii) vitality (perceived level of energy & fatigue); and (iii) potential mediators (such as residential greenness), using pooled and city-level data. Results showed that low levels of childhood NOE exposure were associated with worse mental health in adulthood and not with vitality. No evidence of mediation was found. This article would be of interest to readers of IJERPH and is well written and researched. One drawback from the experimental design was that only 1 question was used to assess retrospective childhood NOE, but this fact was clearly outlined in the Discussion.
Major problems:
none
Minor Problems:
Data for the main finding in Table 3 concerning the association between mental health and childhood NOE exposure is consistent for pooled data and for the cities of Barcelona, Kaunas and Stoke-on-Trent, but there is a clear disconnect for Doetinchem which was not statistically significant. The authors should note this finding and postulate why the results for Doetinchem differ from the other cities.
Authors: While the childhood NOE exposure was significantly associated with mental health on pooled level and for Barcelona, Kaunas, and Stoke-on-Trent, there was no association for Doetinchem respondents. Doetinchem is the city with the highest NDVI values, childhood NOE exposure and the other NOE variables were also highest for participants in Doetinchem. The high NOE exposure for Doetinchem participants (and low variability among them) is probably the reason why no association was seen. The following paragraph has been added:
Line 438-441: “On city level, only for Doetinchem participants no association was found between childhood NOE exposure and mental health. Doetinchem is the greenest city and childhood NOE exposure was high. Low variability in NOE exposure among these participants is the most likely explanation for the missing association.”
By looking at current residential greenness, is the assumption made that childhood conditions were the same, or was the objective to determine whether adulthood residential greenness could alleviate a lack of childhood NOE? Thus, it is not entirely clear how adulthood residential greenness status could ‘mediate’ the association between childhood NOE and mental health.
Authors: There were two main reasons for investigating current residential greenness. Firstly, current residential greeness has been seen to affect current mental health and vitality. Therefore, current residential greeness may act as a potential confounder, which is why we had to adjust for it. Secondly, we hypothesized that childhood NOE might influence the self-selection into greenner neighborhoods (i.e. people with higher levels of CNE exposure could have a preference for greenner living environments) and residential greenness could therefore act as a mediator. This has been included in the manuscript by adjusting the following sentences to the introduction:
Line 89-91: “Furthermore, some authors have suggested that childhood NOE exposure can have long-term influence on perceptions, evaluations and use of NOE, but evidence on this proposed relationship is scarce.”
Line 97-103: “These findings show that childhood NOE exposure could potentially influence the way people interact with NOE in adulthood. Childhood NOE exposure might thus affect the choice for living in an area with a certain amount of residential surrounding greenness in adulthood, and the perceived amount, use, satisfaction, and importance of NOE in adulthood.”
Line 117: “We hypothesized that low levels of childhood NOE exposure would be associated with the choice of a neighborhood with lower residential surrounding greenness, lower perceived amount of NOE, lower use of NOE...”
Potential confounding by residential surrounding greenness (or current exposure to NOE) is described in the discussion (p. 23, lines 481-484):
“It also showed that the association between childhood NOE exposure and mental health was not confounded by current NOE exposure in adulthood. This was an important addition to a previous study that demonstrated associations between childhood NOE exposure and mental health in adulthood, but that did not take the NOE exposure in adulthood into account [43].”
Lines 28-29: first sentence of the Abstract is not a complete sentence.
Authors: The first sentence has been changed and reads now as follows:
“Exposure to natural outdoor environments (NOE) is associated with health benefits, however, evidence on the impact of NOE exposure during childhood on mental health (MH) and vitality in adulthood is scarce.”
Lines 52-53: change “Mental health is negatively influenced…” to “Mental health can be negatively influenced…” and “others exposure” to “other exposures”
Authors: Thank you, we have accepted your suggestion. The sentence is now:
“Mental health can be negatively influenced by urban built environments, including among other, exposures to noise, crowds and lack of green spaces [3-6].”
Lines 54-55: delete “, in contrast to our study,”
Authors: We have deleted this part of the sentence and it is now:
“Previous studies found that green spaces positively influence physical and mental health as well as general well-being in adults [7-13].”
Line 56: define NOE here rather than on line 67
Authors: We have corrected this error and define NOE where it first appears.
Line 69: define BMI
Authors: This has also been corrected.
“In contrast, the availability of NOE and the time that children spend in them have been associated with increased self-esteem, quality of life, respiratory health, physical activity, and lower body mass index (BMI) [31-34]”.
Line 333: change ‘study’ to ‘studies’
Authors: This has been changed. It is now:
“This is one of the first epidemiological studies to show that low levels of childhood NOE exposure are associated with worse mental health in adulthood when compared to those who had high levels of childhood NOE exposure.”
Line 336: delete the first comma
Authors: Thank you, it has been deleted.
Line 414: change “the nature” to “nature”
Authors: Thank you, this has also been changed.
Line 420: change “with” to “which”
Authors: Thank you, this has also been changed.
Line 423: delete the comma
Authors: The comma has been deleted.
Reviewer 3 Report
It was a pleasure to review this article. I appreciate that the authors made very clear the methodological limitations and all interpretation of the data was within the context of those limitations. The article was well written overall. Decisions about data analysis were very clear. It is likely many readers will be qualitative researchers and not overly familiar with regression. Transparency in reporting is also very helpful for researchers familiar with statistical analysis.
Although there are obvious limitations, I believe this is an important paper to trigger further investigation. It gives researchers a step-up to make decisions about the next stages in research and provides a good basis for research funding. Additionally, although the research is retrospective it will be increasingly difficult to conduct prospective studies of this type due to rapid urbanisation and loss of nature occurring globally.
I believe this article will be of interest to a wide range of researchers, policy-makers, practitioners.
My suggestions for improvement are:
Change 'statistically significant' to 'significant' or 'significantly' throughout the results section. It can be assumed that 'significant' = 'statistically significant' and sometimes the expression is awkward making the sentences difficult to read.
First line of the discussion: Change 'study' to 'studies'
Author Response
Reviewer 3
It was a pleasure to review this article. I appreciate that the authors made very clear the methodological limitations and all interpretation of the data was within the context of those limitations. The article was well written overall. Decisions about data analysis were very clear. It is likely many readers will be qualitative researchers and not overly familiar with regression. Transparency in reporting is also very helpful for researchers familiar with statistical analysis.
Although there are obvious limitations, I believe this is an important paper to trigger further investigation. It gives researchers a step-up to make decisions about the next stages in research and provides a good basis for research funding. Additionally, although the research is retrospective it will be increasingly difficult to conduct prospective studies of this type due to rapid urbanisation and loss of nature occurring globally.
I believe this article will be of interest to a wide range of researchers, policy-makers, practitioners.
My suggestions for improvement are:
Change 'statistically significant' to 'significant' or 'significantly' throughout the results section. It can be assumed that 'significant' = 'statistically significant' and sometimes the expression is awkward making the sentences difficult to read.
Authors: We thank you for that advice and have changed all sentences accordingly.
First line of the discussion: Change 'study' to 'studies'
Authors: Thank you, this has also been changed.